# Early exclusive breastfeeding cessation and postpartum depression: Assessing the mediating and moderating role of maternal stress and social support

Md Jahirul Islam[1,2]*, Lisa Broidy[1,3], Kathleen Baird[4], Mosiur Rahman[5], Khondker Mohammad Zobair[6]

**1** Griffith Criminology Institute, Mt Gravatt, Brisbane, Queensland, Australia, **2** Skills for Employment Investment Program (SEIP) Project, Finance Division, Ministry of Finance, Dhaka, Bangladesh, **3** Department of Sociology, University of New Mexico, Albuquerque, NM, United States of America, **4** Faculty of Health, School of Nursing and Midwifery, University of Technology Sydney, Sydney, New South Wales, Australia, **5** Department of Population Science and Human Resource Development, University of Rajshahi, Rajshahi, Bangladesh, **6** Department of International Business and Asian Studies, Griffith University, Brisbane, Queensland, Australia

* mdjahirul.bd@gmail.com

**Data Availability Statement:** All relevant data are within the manuscript and its Supporting Information.

## Abstract

### Background

Early termination of exclusive breastfeeding (EBF) and postpartum depression (PPD) are both recognized as global health problems. Recent literature reviews demonstrate a notable link between PPD and breastfeeding outcomes, however, the underlying mechanisms linking the two remain unclear.

### Objectives

The aim of the study is to: 1) explore the comparative risk for PPD among new mothers who terminated EBF before the 6-month mark, compared to those who did not; and 2) test whether maternal stress and social support operate to mediate and/or moderate the relationship between EBF and PPD.

### Methods

Between October 2015 and January 2016, a cross-sectional study was carried out among 426 new mothers of Bangladesh who were six months postpartum.

### Results

Based on the multivariate logistic regression model, non-exclusively breastfeeding mothers were 7.58-fold more likely to experience PPD (95% CI [3.94, 14.59]) than exclusively breastfeeding mothers. Additionally, maternal stress and social support not only partially mediate the relationship between EBF and PPD but also substantially moderate this relationship. Specifically, the odds of PPD are significantly higher among mothers who had early EBF interruption in conjunction with increased stress levels and limited social support.

**Funding:** The authors received no specific funding for this work.

**Competing interests:** NO authors have competing interests.

## Conclusions

Current evidence suggests that concurrent screening for EBF difficulties and maternal stress are important red flags that might hint at complications even before mother's screen positive for PPD. Support and care from family members can provide assistance in overcoming this issue.

## Introduction

Postpartum depression (PPD) is widely acknowledged as a significant public health issue, affecting 3–19% of women worldwide [1,2]. PPD typically occurs within one month following childbirth with symptoms that may include feelings of extreme sadness, guilt, low self-worth, and hopelessness [3]. Evidence suggests that PPD can lead to long-lasting deleterious effects on both mothers and their children, including impaired mother-infant interaction [2,4,5], parenting stress [6], infant attachment problems [7], maternal death due to suicide [8] and children's poor cognitive and physical development [2,4]. In addition, PPD involves significant economic costs, both in terms of healthcare expenditures and lost work productivity [9]. There is also mounting evidence that it is associated with early interruption of exclusive breastfeeding (EBF) [7,10,11].

The association between PPD and the early interruption or termination of exclusive breast feeding is notable because of the well-established short- and long-term health benefits of breastfeeding. Citing these benefits, the World Health Organization (WHO) strongly recommends EBF with no additional food types other than required medicines, vitamins, and minerals in the first six months of an infant's life [12]. Evidence suggests that exclusively breastfed babies have 15 times greater likelihood of survival in the first six months of life than non-exclusively breastfed children [13]. While the promotion of breastfeeding is a global priority for improving child mortality and morbidity, the prevalence of breastfeeding during the first few weeks postpartum dramatically declines in both developed and developing countries, and EBF during an infant's first six months is rare [14,15]. Given that PPD is a known correlate of early termination of EBF, early identification of mothers at-risk for both PPD and EBF termination is a significant global health priority [16]. However, research has yet to fully articulate the mechanisms that might link PPD with the early cessation of EBF, which complicates efforts to develop targeted interventions with women at risk for PPD and EBF.

Complicating our understanding of the links between EBF and PPD is a lack of clarity around the direction of the relationship between the two. A range of studies have illustrated that early EBF interruption is associated with increased risk of PPD [7,17,18], while other recent studies have indicated that PPD occurs prior to discontinuation of EBF [11,19,20]. At the same time, several studies suggest no association [21–24]; or conversely find that breastfeeding mothers are at higher risk for PPD [25]. Because of the inconsistent and insufficient evidence, the nature of this association and the mechanisms underlying it remain unclear [10,26,27]. It may be that early termination of EBF and PPD only become comorbid under specific conditions. Here, we suggest that social support and maternal stress are among the plausible mechanisms that might account for the link between EBF termination and PPD.

The nexus between EBF and women's postpartum mental health may be driven by a number of psychosocial and biological influences [11]. Earlier studies suggests that breastfeeding mothers appear to be calmer, less anxious, and less stressed [28,29]. Because of increased social and educational awareness regarding the established health benefits of breastfeeding, many

women intend to breastfeed prenatally and also feel an intense social pressure to do so. Mothers experiencing breastfeeding difficulties report culpability and loneliness, which ultimately leads feelings of worthlessness and defeat [29,30]. The psychological displeasure due to the inability to breastfeed increases maternal anxiety and stress levels and other postpartum adjustment issues [18,27,29]. Social support is among the most crucial resources for navigating the stressors of early motherhood, with benefits to both mother and child [31,32]. Those with high social support are less likely to terminate EBF and report less PPD [33,34]. Alternatively, women with limited social support who experience stress and anxiety around EBF are at increased risk of PPD [7,35–37].

Given this literature, we see two possible pathways through which maternal stress and social support might influence the link between EBF and PPD. First, maternal stress and limited social support may be associated with both early termination of EBF and PPD, thereby *mediating* the relationship between the two. In other words, accounting for the association of maternal stress and decreased social support with both EBF and PPD would reduce the relationship between these two postpartum outcomes. In addition to this mediating model in which stress and social support act as intervening mechanisms explaining the link between EBF and PPD, it may also be the case that maternal stress and limited social support exaggerate any link between early termination of EBF and PPD, creating a *moderating* effect. In other words, EBF has a stronger impact on PPD when it interacts with high maternal stress or low social support, thereby moderating the direct relationship between EBF on PPD. Exploring these potential mediation and moderation effects of perceived stress and social support on the link between EBF and PPD is crucial for identifying key pathways and processes, and for improving guidance for the formulation of effective intervention strategies.

Research on the links between EBF and PPD has been predominantly carried out in high-income countries, with varying definitions of EBF. Less is known about the influence of EBF on maternal mental health outcomes in low- and middle-income countries. In the South Asian region, only two recent studies from Pakistan [38] and Bangladesh [11] investigated the influence of PPD on EBF outcomes. In our previous study, it was revealed that mothers who experienced postpartum depressive symptoms exhibited a lower likelihood of EBF than those who reported no such symptoms [11]. However, no studies were found in South Asia exploring how or why there is a significant association between early termination of EBF on the development of PPD. Specifically, we know of no other studies from South Asia that have investigated the influence of maternal stress and social support on the association between EBF and PPD. In response, this study aims to: 1) explore the comparative risk for PPD among new mothers who terminated EBF before the 6-month mark, compared to those who did not; and 2) test whether maternal stress and social support operate to mediate and/or moderate the link between EBF and PPD. We hypothesized that non-exclusive breastfeeding is linked with higher risk of PPD and that maternal perceived stress and limited social support modify the relationship between EBF and PPD.

## Materials and methods

### Study population

The current study is drawn from a much larger project, which mainly investigates maternal and child health before, during and after pregnancy in Bangladesh. This extensive dataset was used to produce several research outcomes exploring the correlates and changing pattern of intimate partner violence (IPV) before, during, and after pregnancy [39,40], the influence of IPV on experiencing PPD [41], maternal healthcare services [42] and suicidal ideation [8], psychosocial factors of EBF [11], and the influence of childhood maltreatment on EBF behaviours

[43]. As described previously [11,41], a population-based cross-sectional survey of new mothers was carried out between October 2015 and January 2016 in two *Upazilas* (sub-districts) of the Chandpur district of Bangladesh. Virtually all mothers in Bangladesh rely on government health clinics for their maternal and child health needs [44]. For this study, four hundred and twenty-six new mothers who visited government-sponsored community immunization clinics to receive vaccinations for their babies constitute the sampling frame. Married women of reproductive age (15–49 years) with a child of six months or younger were included in the study. It is important to note here that, given socio-cultural norms around marriage and parenthood (key markers of adulthood) in Bangladesh, the ethics board determined that, for this study, married women aged 15 or older could agree to participate without parental consent.

## Data collection

Detailed data collection procedures have been reported previously [11,41]. A multistage random sampling design was applied for drawing the sample from selected immunization clinics. Stage by stage, two *Upazilas*, 10 *Unions* (5 *Unions* per *Upazila*), and 80 immunization clinics (8 clinics per *Union*) were chosen randomly to conduct the study. *Union*-wide immunization clinic lists were obtained from the Upazila Health Centre to select immunization clinics. Once *Unions* were chosen randomly, starting with the second immunization clinic on the list, every third clinic was selected for the research.

If eligible to participate, women were invited to participate in a face-to-face survey. Because several survey questions were relatively technical and hard to understand by the respondents with limited literacy, a closed-form interviewer-administered questionnaire was utilized rather than using a self-administered questionnaire. At each of the selected immunization clinics, interviewers approached every woman who visited the clinic and asked each of the eligible mothers to take part in the study. Attaining the desired sample size of 426, a total of 453 women were approached.

## Human participation protection

The project received ethical approval from the National Research Ethics Committee of the Bangladesh Medical Research Council (BMRC/NREC/2013-2016/305) and Griffith University Human Research Ethics Committee (CCJ/41/14/HREC). Taking the cultural context and sensitivity of the study into account, verbal informed consent was taken from every participant after informing them of the purpose of the study, the confidential nature of the interview, and their right to withdraw from the interview at any stage without consequences. As per our approved ethics agreements, verbal informed consent was collected from all participants, but not formally documented in order to ensure respondents' anonymity and to avert any legal consequences.

## Measures

**PPD.** The main outcome variable in the current study was PPD. The Bangla version of the Edinburgh Postpartum Depression Scale (EPDS-B) was applied to ascertain postpartum depressive symptoms among new mothers [45]. The EPDS comprises 10 four-point Likert scale statements (0–3) concerning women's feelings of enjoyment, stress, fear and anxiety (see Appendix for a full list of items). A sum score ranges from 0–30 with higher total scores specifying higher depressive symptoms. The validation of the EPDS-B suggests a cut-off score of $\geq 10$, sensitivity 89% and specificity 87% [46]. Using this cut-off score, postpartum mothers were categorized as either non-depressed (score $< 10 = 0$) or depressed (score $\geq 10 = 1$). The Cronbach's α for this scale in this study was .90.

**EBF.** The breastfeeding practice was classified as EBF using the WHO's definition: "up to six months of age, the infant receives only breast milk without any additional liquids or solids —not even water—except oral rehydration solution, or drops/syrups of vitamins, minerals, or medicines" [47, p.2]. Accordingly, each postpartum mother was asked if she had ever breastfed her infant, if she was continuing breastfeeding, and if so, whether any additional food was fed to her infant in the past [8]. The feeding behaviour was classified as either EBF (= 1) if a baby was breastfed only since birth, or non-EBF (= 0) if a child was fed anything other than breastmilk.

## Maternal perceived stress

Maternal perceived stress was assessed using the 10-item Cohen's Perceived Stress Scale (PSS) on a 0 (never) to 4 (very often) Likert scale [48]. This scale measures perceived stress by asking postpartum mothers about their feelings and thoughts during the previous month in relation to such things as level of stress and the ability to cope with situations and handle personal problems during the last month (see Appendix for a full list of scale items). After reverse scoring for some items, total scores range from 0–40 with higher total scores indicate higher levels of stress. The Bangla version of PSS has been validated in Bangladesh [43]. The PSS does not have predefined cut-off values as it is not a diagnostic instrument [49]. Following a study in Pakistan [50], this study uses 20 as the cut-off score to classify women as either low stress (score <20 = 0) or a high stress (score ≥20 = 1) (Cronbach's $\alpha$ = .93).

## Social support

Social support was assessed using Chan *et al.'s* (2011) 10-item scale [51]. The Likert scale scored as 1 (strongly disagree) to 4 (strongly agree) asking mothers whether they have people they can talk to and rely on (see Appendix for full list of items). A higher score indicates greater self-reported social support. The total score was classified into two groups: the lowest tertile was coded as limited social support (= 0), and the higher two tertles were coded together as high social support (= 1) (Cronbach's $\alpha$ = .90).

## Control variables

A range of control variables were included in this study that have been theoretically and empirically linked to PPD [2,52] and EBF [53,54]. Maternal age was classified into adolescence, young adulthood, and adulthood (14–18 years = 0, 19–24 years = 1, or 25 years and over = 2). Educational level was categorized according to the country's formal education system: no education (0 years = 0), primary (1–5 years = 1), and secondary and higher (6 years or more = 2). Family monthly income was categorized based on the national average (BDT 8500, ~ 109 USD) as ≤ BDT 8500 (= 0) versus >BDT 8500 (= 1). Place of residence was grouped as rural (= 0) versus urban (= 1). Reproductive attributes, such as parity (primiparous = 0, multiparous = 1), pregnancy intention (unintended = 0, intended = 1), mode of birth (caesarean = 0, vaginal = 1), and obstetric complications (no = 0, yes = 1) were also taken into account.

**Prior depression.** Each woman was asked if she had suffered from any of the PPD symptoms around the time of pregnancy, referred to herein as prior depression (categorized as: score <10 = no = 0 and score ≥10 = yes = 1). We control for prior depression since it increases the odds of PPD [41].

**Childhood sexual abuse (CSA).** CSA was determined by any of the following acts before age 15: forced to have sexual intercourse, or forced to touch someone or be touched, kiss, undress or perform any other sexual acts against her will by anyone (0 = no, 1 = yes). We control for CSA since it is associated with maternal stress [55].

## Statistical analysis

Data analysis was performed by using SPSS version 24.0 for Windows (SPSS Inc., Chicago, IL, USA). Descriptive statistics for mothers' socio-demographic, reproductive, and psychosocial characteristics and EBF outcomes were calculated. To assess the distributions of PPD by relevant covariates, bivariate analysis was performed using cross tabulation. The χ2 test was adopted to evaluate the differences in PPD rates by mothers' psychosocial characteristics. We set the level of significance for all analyses at $p < .05$ (two-tailed). The unadjusted associations between the outcome and other variables were assessed through odds ratios and corresponding 95% confidence intervals (CI). The multicollinearity was tested by variance inflation factors (VIF), all of which were below the standard cut-off score of 2.5.

Adjusted odds ratios (AOR) and 95% CI were estimated by using multivariate logistic regression models to compare the strength of the relationship between each of the confounders and PPD. First, three adjusted multivariate logistic regression models were designed—one separate model for early termination of EBF, maternal perceived stress, and social support to evaluate the independent influence of each on PPD (Table 2, Model 1–3). All the confounders were introduced simultaneously into the logistic regression models. Another set of multivariate models were designed to explore whether maternal stress and social support mediate and/ or moderate any association between the early termination of EBF and PPD. In the first occasion (mediation), maternal stress and social support were introduced into the model to assess if the association between EBF termination and PPD was attenuated once maternal stress and social support were taken into consideration (Table 2, Model 4). In the second occasion (moderation), an interaction term for maternal stress and EBF was introduced to investigate whether higher levels of stress exaggerate the association between EBF termination and PPD. Additionally, another interaction term for social support and EBF was introduced to determine whether limited social support exaggerates the association between of EBF termination on PPD.

## Results

### Breastfeeding outcomes

Fig 1 portrays the frequency distribution of various feeding practices among respondents. Within the total sample, 47 women introduced bottle or formula feeding (11.0%), 193 women continued mixed feeding (45.3%), and the remaining 186 women exclusively breastfed their babies at any point until the age of six months (43.7%). Fig 1 also depicts the feeding practice-wise frequency distribution of the incidence of PPD among the respondents. In practice, the incidence of PPD was highest among mixed feeding mothers (58.3%) and lowest among exclusively breastfeeding mothers (8.6%).

Fig 2 depicts the EBF and PPD rates by maternal postpartum age in the sample. The prevalence of EBF was approximately 71.0% in the first month, whereas the rate dramatically declined to 29%in the sixth month. Moreover, 23.0% of new mothers at month 1 and 40.0% of women at month 6 experienced PPD. The figure clearly demonstrates a steady reduction in EBF and a fairly dramatic shift in PPD at about 4 months postpartum.

### General characteristics

Table 1 provides descriptive data for the sample and evaluates differences in rates of PPD as a function of several psychosocial characteristics. Most mothers were between 19 and 24 years of age (43.9%), had received either secondary or higher level of education (67.4%), and were living in rural areas (68.5%) at the time of the survey. A large number of respondent's average

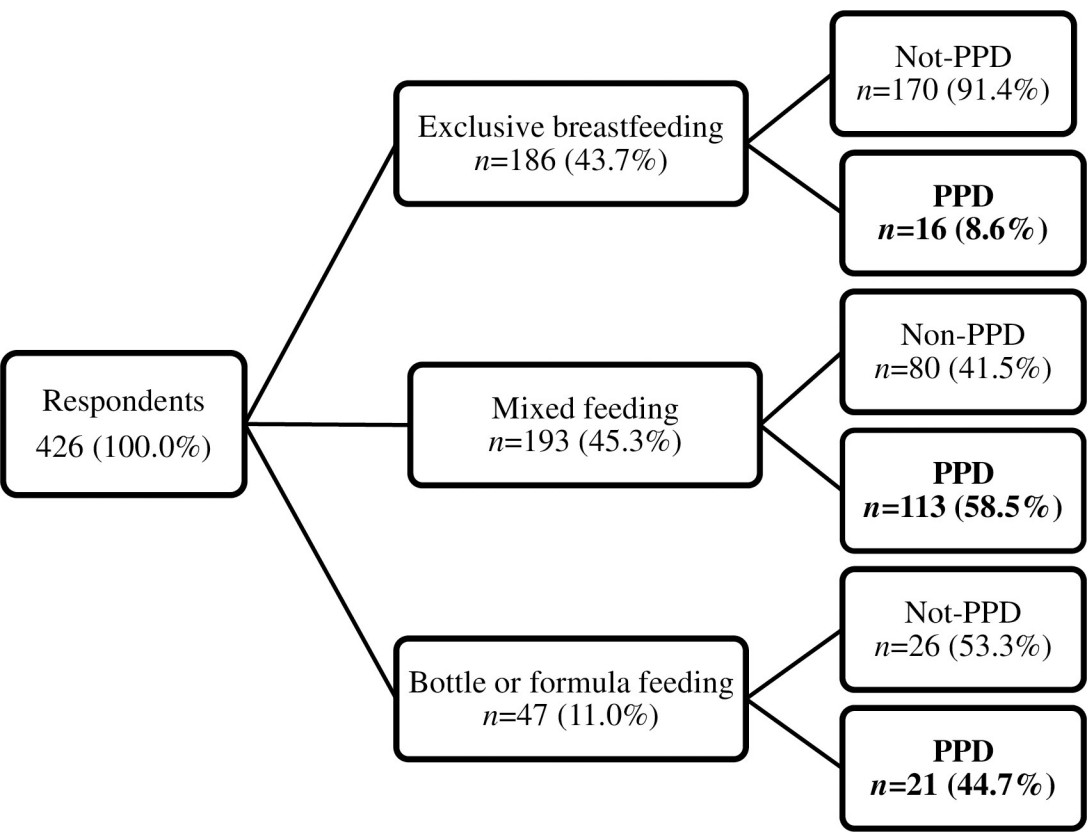

**Fig 1. Frequency distribution of different feeding practices and the incidence of corresponding postpartum depression rate among postpartum women in Bangladesh (N = 426).**

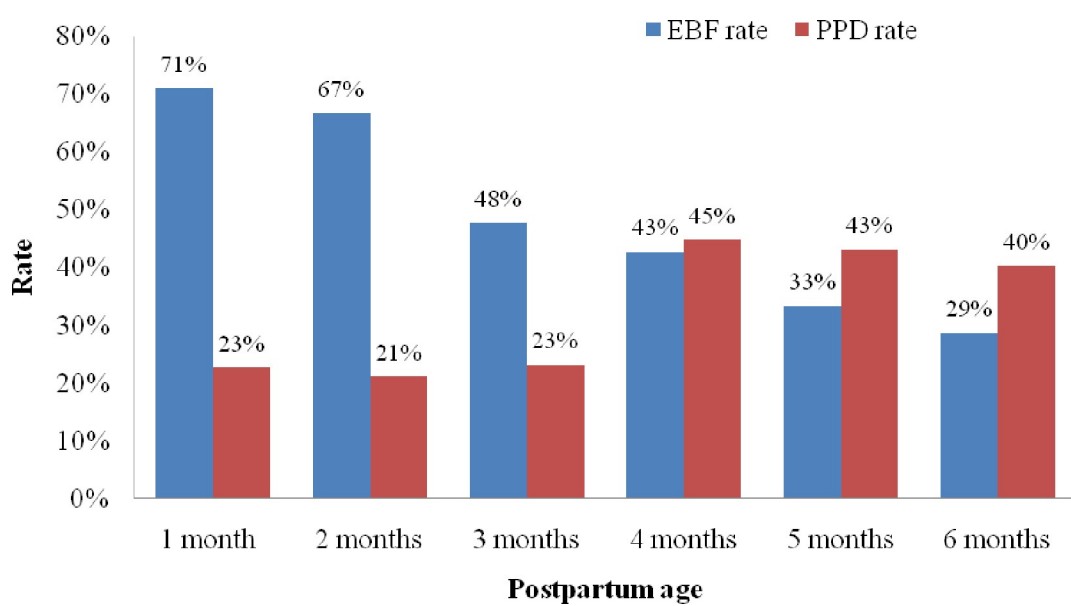

**Fig 2. Rates of exclusive breastfeeding and PPD by maternal postpartum age.**

**Table 1. General characteristics of the participants by the occurrence of postpartum depression and their unadjusted associations (N = 426).**

| Characteristics | n (%) | Postpartum depression | | Crude OR (95% CI) |
|---|---|---|---|---|
| | | (%) | p-value | |
| **Maternal perceived stress** | | | | |
| Low stress | 183 (43.0) | 11.5 | < .001 | 1.00 |
| High stress | 243 (57.0) | 53.1 | | 8.73 (5.19–14.68) * |
| **Social support** | | | | |
| High | 282 (66.2) | 20.2 | < .001 | 1.00 |
| Limited | 144 (33.8) | 64.6 | | 7.20 (4.60–11.27) * |
| **Breastfeeding practices** | | | | |
| Exclusive breastfeeding | 240 (56.3) | 56.8 | < .001 | 1.00 |
| Not exclusive breastfeeding | 186 (43.7) | 8.6 | | 13.43 (7.58–23.81) * |
| **Maternal age** | | | | |
| 14–18 | 106 (24.9) | 36.8 | .35 | 1.00 |
| 19–24 | 187 (43.9) | 31.6 | | 0.79 (0.48–1.31) |
| ≥ 25 | 133 (31.2) | 39.1 | | 1.10 (0.65–1.87) |
| **Maternal education** | | | | |
| No formal education | 35 (8.2) | 62.9 | < .001 | 1.00 |
| Primary | 104 (24.4) | 48.1 | | 0.55 (0.25–1.20) |
| Secondary and higher | 287 (67.4) | 27.2 | | 0.22 (0.11–0.46) * |
| **Family monthly income, BDT** | | | | |
| ≤8500 (~ 109 USD) | 163 (38.3) | 52.1 | < .001 | 1.00 |
| >8500 | 263 (61.7) | 24.7 | | 0.31 (0.20–0.46) * |
| **Place of residence** | | | | |
| Rural | 292 (68.5) | 37.3 | .17 | 1.00 |
| Urban | 134 (31.5) | 30.6 | | 0.74 (0.48–1.15) |
| **Parity** | | | | |
| Primiparous | 175 (41.1) | 30.3 | .08 | 1.00 |
| Multiparous | 251 (58.9) | 38.6 | | 1.45 (0.96–2.19) |
| **Pregnancy intention** | | | | |
| Unintended | 107 (25.1) | 43.9 | .03 | 1.00 |
| Intended | 319 (74.9) | 32.3 | | 0.61 (0.39–0.95) *** |
| **Obstetric complications** | | | | |
| No | 357 (83.8) | 33.9 | .20 | 1,00 |
| Yes | 69 (16.2) | 42.0 | | 1.41 (0.84–2.39) |
| **Mode of childbirth** | | | | |
| Caesarean section | 131 (30.8) | 25.2 | .004 | 1.00 |
| Spontaneous Vaginal | 295 (69.2) | 39.7 | | 1.95 (1.23–3.09) ** |
| **Prior depression** | | | | |
| No | 312 (73.2) | 22.4 | < .001 | 1.00 |
| Yes | 114 (26.8) | 70.2 | | 8.13 (5.03–13.16) * |
| **Childhood sexual abuse** | | | | |
| No | 364 (85.4) | 33.0 | 0.02 | 1.00 |
| Yes | 62 (14.6)) | 48.4 | | 1.91 (1.11–3.28) *** |
| | **Total** | **35.2** | | |

Here

*p< 0.001

**p< 0.01

***p< 0.05.

monthly income was below the national average (38.3%). Regarding reproductive characteristics, around a quarter of the pregnancies were reported as unintended, 16.2% had experienced obstetric complications, and just over two-thirds of women had spontaneous vaginal births (69.2%). Concerning women's mental health, about one-third of them reported to have limited social support (33.8%), and nearly one in nine women reported that they were victims of sexual abuse before 15 years of age (14.6%). While approximately one in four women (26.6%) had previous depressive symptoms, more than one in three women (35.2%) had PPD symptoms.

## Exclusive breastfeeding and postpartum depression: Bivariate associations

Table 1 demonstrates the results of bivariate analysis performed to assess the distributions of PPD by relevant covariates. The experience of PPD was notably more common among respondents who had no formal education, a low income, and limited social support. Postpartum mothers were more likely to experience PPD if they also reported unplanned pregnancies and gave spontaneous birth. As anticipated, postpartum mothers who had not exclusively breastfed were significantly more likely to report PPD. Moreover, mothers who had a history of depression, higher levels of maternal perceived stress, and limited social support showed a significantly greater likelihood of PPD. Findings reported in Table 1 depict unadjusted associations between PPD and a range of relevant covariates. Noticeably, no significant association was observed in the bivariate analysis for maternal age, place of residence, parity, and obstetric complications.

## Exclusive breastfeeding and postpartum depression: Multivariate association

Results reported in Table 2 demonstrate the findings of multivariate analyses examining the link between EBF and PPD controlling for all covariates. First, early termination of EBF was examined (Table 2, Model 1), then social support and maternal stress separately (Table 2, Model 2 & 3). Model 1 demonstrates that EBF was significantly associated with PPD, meaning that women who terminated EBF early had a 9.68-fold higher risk (95% CI [5.16–18.16]) of experiencing PPD than women who continued EBF during the first six months postpartum. The odds of PPD were higher for mothers who experience maternal stress and for those who report limited social support. Whether the association between EBF and PPD was partially mediated by maternal stress and social support was also investigated in this study. Results in Model 4 show that both maternal stress and limited social support significantly elevated the odds of PPD. Most importantly, the inclusion of maternal stress and social support in the model considerably decreased the association between early termination of EBF and PPD. This suggests that part of the reason early termination of EBF is associated with PPD is due to the related influence of both maternal stress and social support. In the full model, women with early cessation of EBF were 6.93-fold more likely (95% CI [3.54, 13.57]) to also experience PPD compared to those who were exclusively breastfeeding. Though not full mediation, this is a notable reduction from the base model that showed early cessation of EBF to increase the odds of PPD by 9.68 (95% CI [5.16–18.16]).

## Moderating effect of maternal stress and social support

In addition to evidence that maternal perceived stress and social support partially mediate the link between early termination of EBF and PPD, we found strong support for an interactive effect. A strong interaction effect between EBF and maternal stress on PPD outcomes was observed in the binary regression model (Fig 3). Overall, women who had high levels of maternal stress and early termination of EBF were 17.22 times more likely to experience PPD (95%

**Table 2. Logistic regression Adjusted odds ratios (AOR) with Confidence Interval (CI) for the relationship between EBF and PPD among postpartum women in Bangladesh (N = 426).**

| Independent variable | Postpartum Depression, AOR (95% CI) | | | |
|---|---|---|---|---|
| | Model 1 | Model 2 | Model 3 | Model 4 (Full model) |
| **Breastfeeding practices** | | | | |
| Exclusive breastfeeding | 1.00 | 1.00 | 1.00 | 1.00 |
| Not exclusive breastfeeding | 9.68 (5.16–18.16) * | 8.49 (4.43–16.26) * | 7.61 (3.95–14.65) * | 6.93 (3.54–13.57) * |
| **Social support** | | | | |
| High | - | 1.00 | - | 1.00 |
| Limited | - | 3.63 (2.04–6.45) * | - | 2.95 (1.63–5.34) * |
| **Maternal perceived stress** | | | | |
| Low stress | - | - | 1.00 | 1.00 |
| High stress | - | - | 4.20 (2.16–8.13) * | 3.41 (1.72–6.78) * |
| -2 log likelihood | 375.15 | 355.53 | 355.63 | 342.64 |
| $R^2$ (Cox & Snell) | .34* | .37* | .37* | .39* |
| $R^2$ (Nagelkerke) | .47* | .51* | .51* | .54* |
| Model $\chi2$ | 177.58 | 197.20 | 197.10 | 210.10 |
| Overall model prediction rate | 78.6% | 82.2% | 81.5% | 82.6% |

Here

$^*p < 0.001$.

**Model 1**: Influence of EBF controlling for age, education, place of residence, family monthly income, parity, pregnancy intention, obstetric complications, mode of childbirth, childhood sexual abuse, and prior depression WITHOUT maternal stress and social support.

**Model 2**: Influence of EBF controlling for the above covariates AND social support WITHOUT maternal stress.

**Model 3**: Influence of EBF controlling for the above covariates AND maternal stress WITHOUT social support.

**Model 4**: Influence of EBF controlling for the above covariates AND maternal stress & social support.

CI [5.97, 49.64]) compared to women who exclusively breastfed and had low levels of maternal stress. Additionally, women who had low levels of maternal stress and did not continue EBF were still 5.43-fold more likely to experience PPD (95% CI [1.71, 17.25]) compared to women who had low levels of maternal stress and continued EBF.

We also found a robust interaction effect between EBF and social support on PPD in the logistic regression model (Fig 4). On balance, women who exclusively breastfed and had limited social support were 18.91 times more likely to experience PPD (95% CI [7.73, 46.29]) compared to women with EBF and high social support.

## Discussion

The primary objective of this study was to begin to unpack the link between early termination of exclusive breastfeeding and PPD among new mothers. In addition to establishing a relationship between EBF termination and PPD, we sought to test whether maternal stress and social support mediate or moderate this relationship. The findings of the current study reinforce mounting evidence that non-exclusive breastfeeding is significantly associated with PPD. Furthermore, we extend this literature by examining the contributions of maternal stress and social support to this relationship. Findings indicate that maternal stress and social support are associated with EBF and PPD and that this partly accounts for significant association between EBF and PPD. Furthermore, this relationship is exaggerated when maternal stress is high and social support is limited. These findings have important implications for prevention and early intervention with maternal mental health and breastfeeding practices.

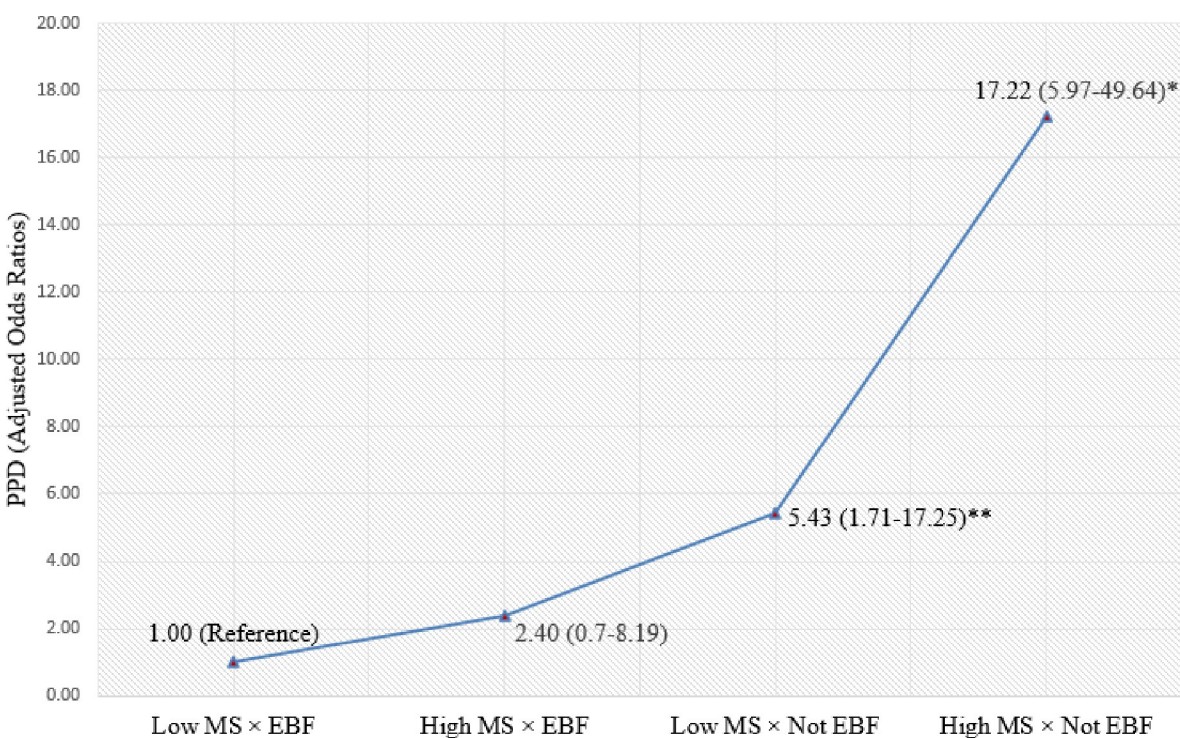

**Fig 3. Combined effect of EBF and maternal perceived stress on postpartum depression.** Controlling for maternal age, maternal education, place of residence, family monthly income, parity, pregnancy intention, obstetrical complications, mode of childbirth, social support, childhood sexual abuse, and previous depressive symptoms; Here $^*p < 0.001$, $^{**}p < 0.01$; MS = Maternal stress; EBF = Exclusive breastfeeding.

The current research enhances the maternal mental health literature by providing evidence of the comorbidity between early cessation of EBF and PPD outcomes, even after adjusting for a range of potential covariates including previous depressive symptoms, obstetric complications, etc. Specifically, even when we account for maternal stress and limited social support during the first 6 months postpartum, mothers who had an early interruption of EBF were 6.93-fold more likely to also experience PPD than those who had exclusively breastfed their infants. In spite of contradictory findings in the literature, particularly on the direction of this association, our results reinforce earlier studies linking early cessation of EBF with PPD [7,17,18,56]. However, important questions of directionality remain and our data, unfortunately, are not fine-grained enough to offer conclusions about the direction of this association.

Beyond the association between EBF cessation and PPD, our results reveal that maternal perceived stress considerably exaggerates these effects. Women who reported early interruption of EBF and *high* levels of maternal stress were significantly more likely to exhibit depressive symptoms than those who exclusively breastfed and had *low* levels of maternal stress. A growing number of studies provide empirical evidence that EBF reduces PPD through several psychological and biological processes directly or indirectly associated with maternal stress [24,57]. Breastfeeding improves maternal psychological well-being not only by regulating maternal-infant sleep and wake patterns [58] but also enhancing mothers' self-efficacy and perceptions of maternal adequacy by improving mothers' emotional involvement with their infant [26] and mother-infant interaction [59]. Exclusive breastfeeding mothers usually sleep an average of 40–45 minutes longer and report less sleep interruption compared to mothers of formula-fed infants [58]. Previous studies show that inadequate or interrupted sleep experienced by new mothers due to formula supplementation during the first few month's

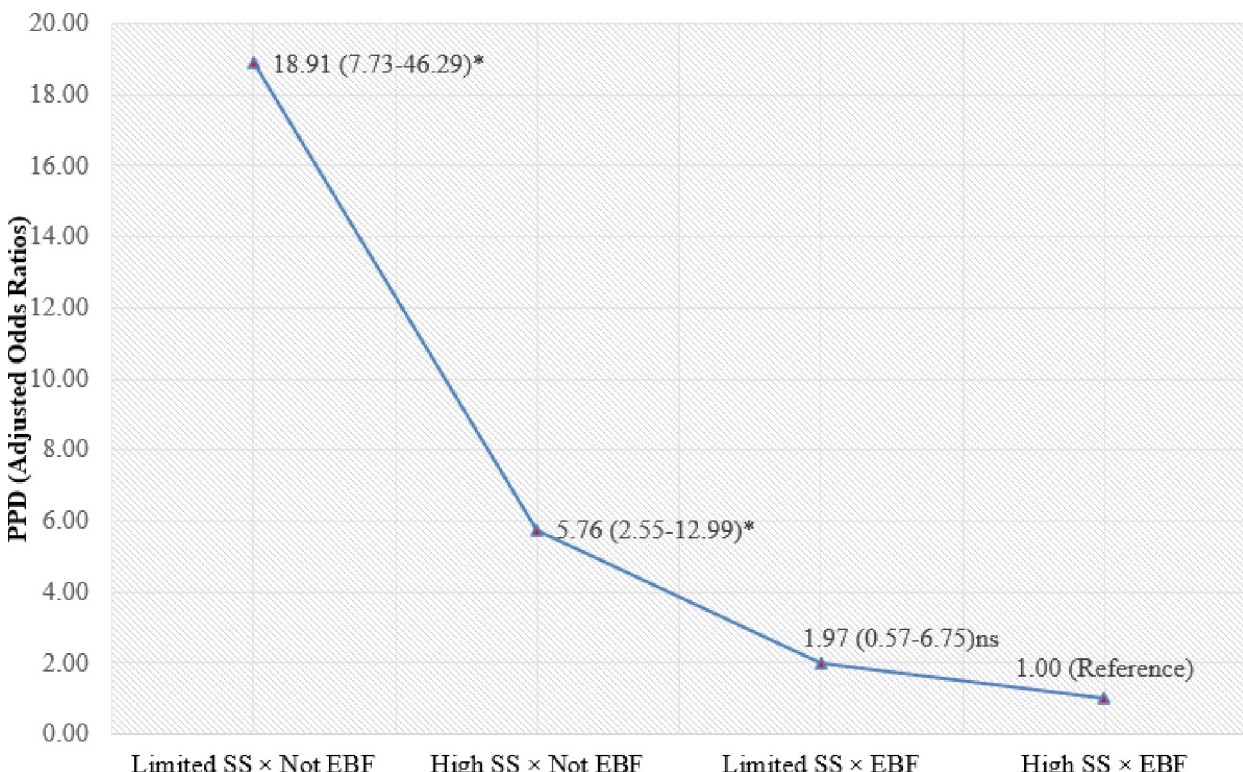

**Fig 4. Combined effect of EBF and social support on postpartum depression.** Controlling for maternal age, maternal education, place of residence, family monthly income, parity, pregnancy intention, obstetric complications, mode of childbirth, childhood sexual abuse, previous depressive Symptoms and maternal stress; Here *$p < 0.001$; SS = Social Support, EBF = Exclusive Breastfeeding.

postpartum leads to increased maternal stress and fatigue [28,60,61]. Consequently, increased levels of maternal stress and anxiety may interfere with the physiological process of milk production or maintaining milk production [28,62,63]. In the delayed lactation process, mothers may erroneously conclude that they are incapable of producing adequate breastmilk, leading to early termination of breastfeeding [63,64]. EBF difficulties may generate feelings of embarrassment and discomfort among women presuming that they will be judged by others for not properly protecting and caring their developing infants, aggravating levels of stress among mothers, which may ultimately contribute to the development of PPD [36,65,66].

An alternative explanation may be that lactation attenuates neuroendocrine and behavioural responses to physical and psychological stress and may act to ameliorate maternal mood [28,67]. In particular, two lactation inducing hormones—oxytocin and prolactin—are reported to have mood-alleviating effects that may promote feelings of nurturance and relaxation [67]. Also, lactation attenuates cortisol stress responses by reducing levels of stress hormone and enhancing sleep [68]. The stress associated with EBF difficulties and sleep deprivation can alter the synthesis and release of prolactin and oxytocin, and interfere with milk secretion and maintaining milk production [62]. This is consistent with evidence suggesting that persistent sleep deprivation and EBF difficulties are significant predictors for the development of PPD [55].

Consistent with other literature [34,41,69], results of this study also implicate social support deficits in the relationship between EBF and PPD. The odds of PPD are significantly higher among mothers with limited social support who terminated exclusive breastfeeding. This is in line with existing literature suggesting that the increased likelihood of EBF is seen among

mothers who report higher social support compared with those who report limited social support [11,70,71]. Previous research shows a linier relationship between greater quality of social support, and stress management and coping [72]. In distress situations, social support buffers the detrimental effects of stressful events by enhancing self-efficacy, leading to fewer depressive symptoms [31,32]. Several studies also have portrayed that quality support from family is significantly connected with better adjustment to and healing from severe diseases, and with healthier neuroendocrine functioning and good mood [73–76]. As such, mothers lacking social support will have a harder time enduring the challenges linked with EBF along with the emotional toll connected with a sense of guilt and inadequacy due to early discontinuation of EBF, exaggerating links between EBF termination and PPD.

Our findings are consistent with other research showing that, compared to mothers without depressive symptoms, mothers with symptoms of postnatal depression are more likely to provide the supplementary formula to their infants in the first year of life [77]. In a large study of women evaluated between 8 and 12 weeks postpartum, it has been established that exclusively breastfeeding mothers had lower levels of depressive symptoms compared to partial breastfeeding mothers [17]. This finding is supported by Ystrom (2012), who also found that, at six months postpartum, both partially breastfeeding as well as exclusively bottle-feeding were associated with higher levels of depressive symptoms in postpartum women when compared to those who solely breastfeed their infants [18]. While our data do not allow us to interrogate the causal mechanisms at play here, we can speculate links between the high levels of depression symptoms and poor feeding outcomes that future research can explore. Postpartum depression might cause negative effects on maternal self-esteem and cognition, which can interrupt breast feeding intentions [77]. Furthermore, women with depressive symptoms may not engage or interact with their baby, they may spend less time experiencing skin—to—skin, not enjoy touching their baby, which in turn may affect their lactation and decrease their milk supply which leads to an increase in their lack of confidence in their own ability to breastfeed. This can then lead to further dissatisfaction and disappointment in relation to their infant feeding practices and increases the likelihood that they rely more on formula feeding and less on breast feeding.

Various professional organizations recommend screening for depression during perinatal and postpartum healthcare visits [78]. However, our work suggests that concurrent screening for problems related to EBF, parenting stress, and social support deficits are also important red flags that might hint at complications even before mother's screen positive for PPD. Screening for these issues may occur early postpartum, at 2–3 months, and finally at 4–6 months. Of course, it remains an important empirical question to determine the most appropriate window for screening and intervention. In this regard, evidence-based policies and guidelines need to be developed by professional organisations, and specialised training should be introduced for midwives, obstetricians, and family workers to identify mothers with EBF difficulties, stress exposure, social disengagement, and depressive symptoms and offer proper support to them. Mother-infant psychotherapy [79], cognitive-behavioural therapy [80], and a 'Nurse-Family Partnership program' or 'Home visitation program' [81–83] were found to be effective in improving maternal mental health as along with family functioning and related social supports. In low- and middle-income countries, well-established referral pathways and support organisations need to be set up before introducing a screening program [84].

## Strengths and limitations

The important features of this particular research are the large sample size ($N$ = 426) and the inclusion of a range of key confounders. This was the first known research of its kind in South

Asian region to explore the association between EBF cessation and PPD in a community-based sample of postpartum mothers. Moreover, it is one of the few studies to assess the mediating and moderating influences of maternal perceived stress on the relationship between EBF and PPD outcomes. Surveying mothers within six months postpartum helped us to significantly minimize the potential recall bias regarding their experience of EBF and maternal mental health. Most importantly, the study results are internationally comparable because of the application of widely recognized standardized instruments for this study. Despite these strengths, the results from this study need to be interpreted in light of several limitations. First, the cross-sectional design of this study restricts us from establishing the exact temporal association between the discontinuation of EBF and PPD outcomes. The current study demonstrates only the strong association between these two significant public health issues. Second, an over-estimation of the rate of EBF may result due to the inclusion of women of postpartum ages between one and six months, because the EBF prevalence is significantly high in the first couple of postpartum months, and then gradually declines. Third, information was obtained through self-report which is subject to some biases. However, the potential recall bias of the respondents is minimized as they were surveyed within six months postpartum. Finally, limited time and budget restrict us from collecting data regarding several confounders, such as the family history of axis I or II disorders, maternal medications use, gestational age, health status of the infant, and difficult infant temperament. Despite such limitations, the research is timely as this particular research adds to the extant literature on the influences of EBF on PPD in low- and middle-income countries.

## Conclusion

Despite high profile campaigns to raise awareness of the importance of EBF and to promote the social acceptability of breastfeeding in Bangladesh, the rate of EBF was found to be only 32% at 4–5 months postpartum [85]. The low prevalence of EBF and its relationship with maternal mental health outcomes underscore the urgency of recognizing both of them as a notable public health issue in Bangladesh. This study reinforces the necessity of early detection and effective treatment of depressed mothers with EBF complications, stress exposure, or limited social support not only to offer need-based support but also to improve EBF outcomes. Since the pathway of the relationship between EBF and PPD remains unclear, longitudinal studies assessing the direction of the association between PPD outcomes and breastfeeding are needed to get a relatively better understanding of the dynamics and process.

## Supporting information

**S1 Data.**
(SAV)

**S1 Appendix.**
(DOCX)

## Author Contributions

**Conceptualization:** Md Jahirul Islam.

**Data curation:** Md Jahirul Islam, Lisa Broidy.

**Formal analysis:** Md Jahirul Islam.

**Investigation:** Md Jahirul Islam.

**Methodology:** Md Jahirul Islam.

**Project administration:** Md Jahirul Islam.

**Supervision:** Lisa Broidy, Kathleen Baird.

**Visualization:** Md Jahirul Islam.

**Writing – original draft:** Md Jahirul Islam.

**Writing – review & editing:** Md Jahirul Islam, Lisa Broidy, Kathleen Baird, Mosiur Rahman, Khondker Mohammad Zobair.

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
