## [Decision Letter · Decision Letter 0]

9 Sep 2020

PONE-D-20-25535

Early exclusive breastfeeding cessation and postpartum depression: assessing the mediating and moderating role of maternal stress and social support

PLOS ONE

Dear Dr. Islam,

Thank you for submitting your manuscript to PLOS ONE. After careful consideration, we feel that it has merit but does not fully meet PLOS ONE’s publication criteria as it currently stands. Therefore, we invite you to submit a revised version of the manuscript that addresses the points raised during the review process.

We look forward to receiving your revised manuscript.

Kind regards,

Yukiko Washio, Ph.D.

Academic Editor

PLOS ONE

Journal Requirements:

2. Please amend your current ethics statement to address the following concerns: Please explain why written consent was not obtained, how you recorded/documented participant consent, and if the ethics committees/IRBs approved this consent procedure.

3. You indicated that you had ethical approval for your study. In your Methods section, please ensure you have also stated whether you obtained consent from parents or guardians of the minors included in the study or whether the research ethics committee or IRB specifically waived the need for parental consent.

"No"

5. We noted in your submission details that a portion of your manuscript may have been presented or published elsewhere.

"It is an outcome of a large project. Some of the variables of this manuscript are similar to my previously published articles."

Please clarify whether this publication was peer-reviewed and formally published. If this work was previously peer-reviewed and published, in the cover letter please provide the reason that this work does not constitute dual publication and should be included in the current manuscript.

Reviewers' comments:

Reviewer's Responses to Questions

**Comments to the Author**

1. Is the manuscript technically sound, and do the data support the conclusions?

Reviewer #1: Yes

Reviewer #2: Yes

2. Has the statistical analysis been performed appropriately and rigorously? 

Reviewer #1: Yes

Reviewer #2: Yes

3. Have the authors made all data underlying the findings in their manuscript fully available?

Reviewer #1: Yes

Reviewer #2: Yes

4. Is the manuscript presented in an intelligible fashion and written in standard English?

Reviewer #1: Yes

Reviewer #2: Yes

5. Review Comments to the Author

Reviewer #1: Very nice manuscript that does add to the literature on this topic.

I appreciate your discussion about the varied findings of the relationship between PPD and early termination of EPF at the bottom of page 3/top of page 4.

I think it would be helpful to provide a bit more detail about the difference between mediating and moderating effects.

Reviewer #2: This is an exceptionally well-written, well-designed and optimally referenced paper and should be published, requiring only minimal edits.

While recognizing the weaknesses of employing exclusive use of survey data, somewhat temporally removed from the outcome of interest and the limitations of a cross-sectional design, the “n” of over 400 women is meaningful. The highly significant increases in depression recorded among women who stop exclusive breastfeeding early may warrant a closer look at this population when screening for postpartum depression.

It is of interest that more than 25% of women reported either a past history of depressive symptoms or early sexual abuse; these clearly represent a high-risk group.

Not entirely surprising, although an important observation, was the strong association of postpartum depression with an increase in reported levels of stress and the lack of social support.

The paper would be significantly strengthened by the authors having access to the medical record or to have minimally questioned women on the clinical outcomes of their baby at delivery and since the time of birth, as this would be an important confounder.

The first sentence citing PPD rates in the conclusion section is confusing and should be rewritten.

This paper suggests that the screening for depression might occur early, at 2-3 months, and again at 4-6 months, although this screening interval would require validation.

In all, this study from Bangladesh imparts new and important information and should stimulate additional research in new techniques and methodology for screening of postpartum depression. It reinforces the deed for social support and indicates the importance of additionally focusing on women who stop exclusive breastfeeding early.

Identification of postpartum depression begs the question of limited treatment options available to women in under-resourced countries.

6. PLOS authors have the option to publish the peer review history of their article (what does this mean?). If published, this will include your full peer review and any attached files.

Reviewer #1: No

Reviewer #2: No

---

## [Author Response · Author response to Decision Letter 0]

13 Jan 2021

Comment 1: During our internal evaluation of the manuscript, we noted that your study utilizes existing data collected in 2015 for a previously published study, and that you did not perform any additional experiments or collect new data for the current study.

Response: It is clearly evidenced that a misunderstanding regarding our statement of the dataset has arisen. We mentioned that the current manuscript uses existing data collected in 2015 for a large research project that focused on the links between pregnancy, intimate partner violence, and maternal health and wellbeing outcomes. The project consists six research questions/objectives. It is clearly imaginable how large was the magnitude of the study. With each of the objectives, we have published/prepared one separate manuscript using the relevant portion of the same dataset. We have never before used these data to examine the links between the early termination of exclusive breastfeeding and postpartum depression. We confirm this work has not been previously published anywhere. However, in the method and result sections, there might have been little similarities due to the common methodology used in the data collection process and use of some of the common socio-demographic variables, e.g. age, education, income, parity, etc. 

Comment 2: We would like to make you aware that copying extracts from previous publications, especially outside the methods section, word-for-word is unacceptable. In addition, the reproduction of text from published reports has implications for the copyright that may apply to the publications.

Response: We are fully aware of the editor’s concern. Because of the use of the common methodology and some of the similar socio-demographic variables, it may show some similarities with the previous articles. However, we have extensively revised the aforementioned sections of the current manuscript to address the issue. 

Comment 3: Please revise the manuscript to ensure that any portions of the manuscript that have been previously published are fully referenced and cited; rewritten or rephrased; and, if applicable, that permission for republication is obtained and uploaded alongside your submission. 

Response:

We have rewritten and rephrased some of the more generic sections of the manuscript where we found overlaps with our other papers. Additionally, we have referenced and cited our previously published articles (pages 5 & 6). We are not going to republish any articles as each of the manuscripts has a distinct objective. So, copyright permission is not needed.

Comment 4: We ask that you also provide details as to how the current manuscript advances on previous work. Please note that further consideration is dependent on the submission of a manuscript that addresses these concerns about the overlap in text with published work.

Response:

We have added more detail to clarify how the current manuscript advances on previous work (See page 5). Overall, though, this manuscript is fundamentally distinct from others using this large dataset and we are confident it offers important new insights about the relationship between early termination of breastfeeding and postpartum depression. 

We appreciate the opportunity to revise and strengthen the manuscript.

---

## [Decision Letter · Decision Letter 1]

20 Apr 2021

PONE-D-20-25535R1

Early exclusive breastfeeding cessation and postpartum depression: assessing the mediating and moderating role of maternal stress and social support

PLOS ONE

Dear Dr. Islam,

Thank you for submitting your manuscript to PLOS ONE. After careful consideration, we feel that it has merit but does not fully meet PLOS ONE’s publication criteria as it currently stands. Therefore, we invite you to submit a revised version of the manuscript that addresses the points raised during the review process.

We look forward to receiving your revised manuscript.

Kind regards,

Yukiko Washio, Ph.D.

Academic Editor

PLOS ONE

Journal Requirements:

Reviewers' comments:

Reviewer's Responses to Questions

**Comments to the Author**

1. If the authors have adequately addressed your comments raised in a previous round of review and you feel that this manuscript is now acceptable for publication, you may indicate that here to bypass the “Comments to the Author” section, enter your conflict of interest statement in the “Confidential to Editor” section, and submit your "Accept" recommendation.

Reviewer #2: All comments have been addressed

Reviewer #3: All comments have been addressed

2. Is the manuscript technically sound, and do the data support the conclusions?

Reviewer #2: Partly

Reviewer #3: Yes

3. Has the statistical analysis been performed appropriately and rigorously? 

Reviewer #2: Yes

Reviewer #3: Yes

4. Have the authors made all data underlying the findings in their manuscript fully available?

Reviewer #2: Yes

Reviewer #3: Yes

5. Is the manuscript presented in an intelligible fashion and written in standard English?

Reviewer #2: Yes

Reviewer #3: Yes

6. Review Comments to the Author

Reviewer #2: The authors appear to have addressed prior concerns of the reviewers, many of which were focused on being certain that the paper’s conclusions were not a duplicate of data already published from the larger trial.

The weaknesses inherent in a survey-based cross-sectional study are not sufficiently addressed. However, the authors do note the importance of “directionality” (i.e. cause/effect of exclusive breastfeeding on subsequent rates of postpartum depression).

It would be meaningful if the authors were to specifically comment on why women who employ a combination of breast along with bottle feeding had higher rates of PPD as compared to those who exclusively utilized bottle feeding and infant formula.

Despite some limitations of the manuscript, the importance of stress and family support show strong associations with both breastfeeding and postpartum depression. The need to recognize and screen for PPD among women who do not practice EBF for 6 months, whatever the reason, appears to be compelling.

Reviewer #3: The manuscript does not mention whether gestational age was considered in the evaluation, which it appears to not have. If not, this should be mentioned as potential limitation of the study as well as fact that it was self-report.

7. PLOS authors have the option to publish the peer review history of their article (what does this mean?). If published, this will include your full peer review and any attached files.

Reviewer #2: No

Reviewer #3: No

---

## [Author Response · Author response to Decision Letter 1]

24 Apr 2021

We appreciate the generally positive reviews and the insightful comments made by reviewers. We have revised our manuscript in response to editor and reviewers’ comments and have detailed the responses below. We believe the revised manuscript addresses the concerns raised by the editor and reviewers, improving the clarity and overall strength of the document.

Reply to Editor and Reviewer comments 

Editorial Office 

Comment: While revising your submission, please upload your figure files to the Preflight Analysis and Conversion Engine (PACE) digital diagnostic tool, https://pacev2.apexcovantage.com/. 

Response: We have uploaded figure files following the recommendation.

Reviewer 2:

Comment 1: The authors appear to have addressed prior concerns of the reviewers, many of which were focused on being certain that the paper’s conclusions were not a duplicate of data already published from the larger trial.

Response: Thank you very much for the positive decisions regarding the issue raised for the dataset.

Comment 2: The weaknesses inherent in a survey-based cross-sectional study are not sufficiently addressed. However, the authors do note the importance of “directionality” (i.e., cause/effect of exclusive breastfeeding on subsequent rates of postpartum depression).

Response: The issue of a survey-based cross-sectional study has been addressed in page 21.

Comment 3: It would be meaningful if the authors were to specifically comment on why women who employ a combination of breast along with bottle feeding had higher rates of PPD as compared to those who exclusively utilized bottle feeding and infant formula.

Response: Postpartum depression might cause negative effects on maternal self-esteem and cognition and have a detrimental effect on their belief that they can breastfeed their baby. PPD can also lead to a lack of interactions with their baby, such as less skin – to skin, less touching, which can further increase their lack of confidence in their own ability to breastfeed their babies, and lead decreased satisfaction related to their infant feeding practices and further increase in depressive symptoms and increase the need for infant formula. This association has been added into the manuscript (page 19-20).

Comment 4: Despite some limitations of the manuscript, the importance of stress and family support show strong associations with both breastfeeding and postpartum depression. The need to recognize and screen for PPD among women who do not practice EBF for 6 months, whatever the reason, appears to be compelling.

Response: We appreciate the positive concluding remarks regarding the manuscript.

Reviewer 3:

Comment 1: The manuscript does not mention whether gestational age was considered in the evaluation, which it appears to not have. If not, this should be mentioned as potential limitation of the study as well as fact that it was self-report.

Response: Gestational age was not considered in the evaluation which has been addressed as a potential limitation of the study. We have also addressed the issue of self-report (See page 21). 

We appreciate the opportunity to revise and strengthen the manuscript.

---

## [Editor Report · Decision Letter 2]

27 Apr 2021

Early exclusive breastfeeding cessation and postpartum depression: assessing the mediating and moderating role of maternal stress and social support

PONE-D-20-25535R2

Dear Dr. Islam,

We’re pleased to inform you that your manuscript has been judged scientifically suitable for publication and will be formally accepted for publication once it meets all outstanding technical requirements.

Kind regards,

Yukiko Washio, Ph.D.

Academic Editor

PLOS ONE
---

## [Editor Report · Acceptance letter]

29 Apr 2021

PONE-D-20-25535R2 

Early exclusive breastfeeding cessation and postpartum depression: assessing the mediating and moderating role of maternal stress and social support 

Dear Dr. Islam:

I'm pleased to inform you that your manuscript has been deemed suitable for publication in PLOS ONE. Congratulations! Your manuscript is now with our production department. 

Kind regards, 

on behalf of

Dr. Yukiko Washio 

Academic Editor

PLOS ONE